# Design of Experiment to Determine the Effect of the Geometric Variables on Tensile Properties of Carbon Fiber Reinforced Polymer Composites

Joseph C. Janicki, Matthew C. Egloff, Dilpreet S. Bajwa *, Roberta Amendola, Cecily A. Ryan
and Douglas S. Cairns

Mechanical and Industrial Engineering Department, Montana State University, Bozeman, MT 59717, USA
* Correspondence: dilpreet.bajwa@montana.edu; Tel.: +1-406-994-6783

**Abstract:** Carbon fiber reinforced polymers (CFRPs) are increasingly used in the aerospace industry because of their robust mechanical properties and strength to weight ratio. A significant drawback of CFRPs is their resistance to formability when drawing continuous CFRPs into complex shapes as it tends to bridge, resulting in various defects in the final product. However, CFRP made from Stretch Broken Carbon Fiber (SBCF) aims to solve this issue by demonstrating superior formability compared to conventional continuous CFRPs. To study and validate the performance of SBCF, a statistical design of the experiment was conducted using three different types of CFRPs in tow/tape form. Hexcel (Stamford, CT, USA) IM7-G continuous carbon fiber impregnated with Huntsman (The Woodlands, TX, USA) RDM 2019-053 resin system, Hexcel SBCF impregnated with RDM2019-053 resin, and Montana State University manufactured SBCF impregnated with Huntsman RDM 2019-053 resin were tested in a multitude of forming trials and the data were analyzed using a statistical model to evaluate the forming behavior of each fiber type. The results show that for continuous fiber CFRP tows forming, Fmax and Δmax do not show statistical significance based on temperature fluctuations; however, in SBCF CFRP tows forming, Fmax and Δmax is dominated by the temperature and geometry has a low statistical influence on the Fmax. The lower dependence on tool geometry at higher temperatures indicates possibly superior formability of MSU SBCF. Overall findings from this research help define practical testing methods to compare different CFRPs and provide a repeatable approach to creating a statistical model for measuring results from the formability trials.

**Keywords:** carbon fiber; formability; stretch broken carbon fiber; pre-preg; forming test fixture; mechanical properties; statistical modeling

## 1. Introduction

Manufacturing cost for processing carbon fiber reinforced polymers (CFRPs) is one of the main reasons for the limited use of carbon fiber in modern mechanical components despite its superior properties compared to traditional metallics and other fiber composites [1]. Recent advances in the production of carbon fiber (CF) filaments aim to bridge the gap between the mechanical and environmental advantages of CF, and the relative ease of manufacturing for metallic components. Stretch broken carbon fiber (SBCF) has proven more formable than traditional CF while maintaining superior mechanical properties compared to metallic parts [2]. SBCF is produced by utilizing standard continuous fiber tows and stretching it at a pre-determined length using differentially driven rollers so that the fiber breaks at its natural flaws. This process creates an aligned discontinuous tow that can be sized and impregnated by resin into a ply similar to traditional CF pre-preg. The primary benefit of the discontinuous fiber is that, as the temperature approaches the cross-linking temperature of the resin system, the fibers can slide past each other when pressure is applied [3]. As a result of this phenomenon, when a laminate is laid into a

complex mold, the stretch-broken fibers are free to follow the contours of a part's shape more effectively than continuous fibers due to the latter's low strain to failure rate (~2%).

Mechanical properties for uncured CFRP tows are not widely reported, primarily because the cured laminate is of most interest to designers for structural applications. For mesoscale testing of formability at tow level, the mechanical characteristics of both the resin matrix and the carbon fiber are essential because they directly affect the yield stress and yield strain, and both contribute to CFRP formability. Formability is defined by the ease with which a material can be formed while still maintaining its pre-formed properties [4]. The strength of uncured SBCF CFRPs has shown high dependence on temperature, so mechanical property data on uncured laminated tows are of limited use for characterizing formability in this test.

To successfully supplant metallics as the premier material of choice for numerous applications, SBCF must be proven to be easily formable so that manufacturing costs are reduced sufficiently enough to make the final product cost-effective. Commonly referred to as plastic deformation, it is known that ductile materials are usually better suited for forming than brittle crystalline materials such as CF [5]. Since SBCF is discontinuous in nature, it acts as a pseudo-ductile material before curing; therefore, superior forming is expected. Due to the recent and ongoing development of this material and testing equipment and methods of characterization, only limited studies have been conducted to evaluate this behavior of SBCF [1,2]. Montana State University (MSU) has recently designed and implemented a forming fixture that subjects impregnated CF tows into a state of stress termed stretch forming to evaluate the load response of different CF formulations under various geometrical and environmental conditions. The primary objective of this study was to understand the behavior of different types of uncured CFRP and develop a statistical relationship between test forming fixture variables and peak load and displacement values of CF. This relationship is hypothesized to allow future determinations of forming metrics without testing samples physically.

## 2. Materials and Methods

### 2.1. Mechanical Properties of IM7 Carbon Fiber

The CFRPs used in this study were manufactured from a Hexcel carbon fiber base feed stock IM7 CF. IM7 is a high-performance CF that is approved for military aerospace applications. The distinguishing factor between high-performance CF and standard grade CF is the manufacturing controls that are in place during the production of the CF. During the production of high-performance CF, environmental conditions are tightly monitored as well as the tooling conditions. This results in a CF with tight specifications with minimal inherent flaws before being impregnated and cured into the final component [6]. At room temperature (RT), the mechanical properties of the individual CF filaments in a SBCF tow are of little importance because the fiber is primarily discontinuous past the break length of the fiber. The strength of unimpregnated SBCF tows at RT is derived from the interfacial interactions of the filaments and the sizing treatment applied to the fiber [3]. The mechanical properties of continuous Hexcel IM7 CF bundled in a 12,000 fiber (12K) tow are shown in Table 1.

**Table 1.** Mechanical properties of Hexcel 12K IM7 [7].

|  | IM7-12K |
| --- | --- |
| Tensile Strength (GPa) | 5.7 |
| Tensile Modulus (GPa) | 275.8 |
| Elongation at failure | 1.8% |
| Density (g/cm$^3$) | 1.8 |

### 2.2. Properties of RDM2019-053 Resin

Huntsman RDM2019-053 (Huntsman, Corp. The Woodlands, TX, USA) is a proprietary resin system, and chemical and mechanical properties are not publicly available. For

formability testing, resin viscosity is an important factor as it affects the shear strength of the resin and the yield strength of the sample during stretch forming. The resin viscosity values of RDM 2019-053 at relevant temperatures [8] are listed in Table 2.

**Table 2.** Huntsman RDM 2019-053 resin viscosity at selected temperatures.

|  | Temperature (°C) | Viscosity (Pa·s) [9] |
|---|---|---|
| 1 | 21 (70 °F) | >3500 |
| 2 | 57 (135 °F) | 791 |
| 3 | 82 (180 °F) | 168 |
| 4 | 107 (225 °F) | 25 |
| 5 | 121 (250 °F) | 11 |

Testing was conducted between 21 °C (70 °F) and 82 °C (180 °F). Manufacturer's published data show that as the temperature changes from 21 °C (70 °F) to 82 °C (180 °F) the viscosity decreases almost 20 times. This reduction in viscosity implies that temperature will potentially influence the stretch bending strength of the CFRP tows.

*2.3. Sizing Treatment on Carbon Fibers*

A surface treatment termed "sizing" is commonly applied on CF to improve adhesion and surface wettability of the CF tow [10]. As shown in Figure 1, sizing is known to increase the adhesion between the matrix and the CF as well as improve the handleability of the CF [11]. Handleability refers to the fiber protection during processing, alignment of the fiber filaments in the tow, and wettability of the fiber when it comes in contact with the resin system [12]. Hexcel employed two sizing applications for the fibers used in this study. G and GP sizing applications are closely held proprietary formulations, but both are comprised of an uncured epoxy oligomer, phenolic, polyurethane, and vinyl ester compounds with specifically chosen additives that improve adhesion and handleability [13]. GP is known to have comparable properties to G-sized fiber but was produced to have less deleterious effects on the environment.

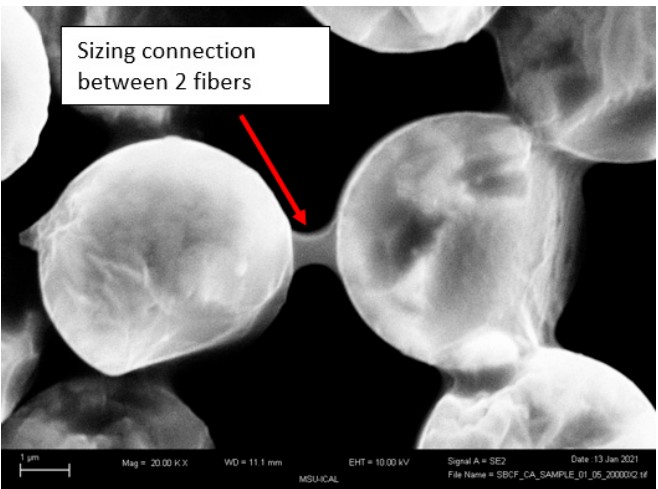

**Figure 1.** FE-SEM microscopy image of sizing bridge between carbon fiber filaments.

*2.4. CF Sample Preparation*

Three types of CF were used in the design of experiment (DOE). All raw carbon fibers were produced by Hexcel Corporation (Stamford, CT, USA) [14]. Fibers and their makeup are displayed in Table 3.

**Table 3.** Carbon Fiber Types Used in DOE.

| Fiber Name | Manufacturer | Fiber Precursor | Sizing Type |
|---|---|---|---|
| 12K Hexcel IM7-GP Cont. | Hexcel | NA | GP |
| Hexcel SBCF | Hexcel | 12K Hexcel IM7 | GP |
| MSU SBCF | MSU | 12K Hexcel IM7 | G |

The specified break length of Hexcel SBCF was 50.8 mm (2 in), which was independently verified by MSU laboratory in a fiber pull-out test [15]. From fiber pullout testing, it has been observed that long corollary fibers up to ~228 mm (9 in) can be found, suggesting higher loads are possible and more variation could be seen throughout Hexcel SBCF. MSU SBCF was manufactured on a modified Hexcel SB2 machine using Hexcel IM7-G continuous fiber as feed stock. G-sized fiber was used because the MSU processing team found that the stretch-breaking process was more effective with G-sizing than GP. The specified break-length of MSU SBCF was 35 mm (1.375 in) which was set from the break distance between the nip rollers on the stretch breaking machine. Each fiber tow was laminated with proprietary Huntsman (The Woodland, TX, USA) RDM2019-053 resin film using the Pro-Lam (Hamilton, MO, USA) in the MSU composites laboratory.

### 2.5. Fiber Volume Content of Impregnated Tow Samples

During the CFRPs manufacturing process, the "dry" CF tow is processed through a lamination machine where resin film is compacted at elevated temperature and pressure to impregnate the CF with the desired resin matrix. Using MSU lamination techniques (see sample preparation), three different fiber types pre-pregs were manufactured using RDM2019-053 resin. An essential factor in the commercial production of pre-impregnated CFRPs is the fiber to resin ratio, or simply fiber volume ratio, which was calculated using Equation (1).

$$V_f = \frac{v_f}{v_c} \tag{1}$$

where $V_f$ is the fiber volume ratio, $v_f$ is the volume of fiber, and $v_c$ is the volume of the composite. For commercial pre-pregs, the industry standard for fiber volume is typically between 50% to 65% depending on the application [16]. The solvent wash process was utilized by MSU to determine the fiber volume. It consisted of washing the uncured pre-preg tow in a 50–50 bath of n-methyl-pyrrolidone (NMP) and acetone solution. The solvent can strip the resin away from the fiber after repeated cycles, leaving only the fiber, which can be weighed to determine the final fiber volume [17].

### 2.6. Test Specimen Fabrication

The fiber reinforced polymer composite tow samples were prepared using Hexcel continuous IM7-G, Hexcel SBCF, and MSU SBCF carbon fiber impregnated with Huntsman RDM 2019-053 resin. Before resin impregnation, precautions were taken to ensure all fibers were aligned correctly, free of twists or misalignment. The average carbon fiber volume fraction for a tow sample was estimated to be 50% ± 2%. After the tow was impregnated with the resin, it was prepared for placement on CNC-routed cardstock tabs (32 mm × 19 mm). The detailed procedure for manufacturing test samples has been published in a previous study [18]. If the specimens were not tested immediately, they were stored in a freezer at −18 °C to prevent the resin from cross-linking [9]. Test samples measured 203 mm between the inside of the tabs, and 7.6 mm wide and 0.25 mm thick. Figure 2 is an illustration of tow sample fabrication.

### 2.7. Experimental Design

The technique for testing the formability characteristics of CF in the DOE was based on using the MSU designed forming fixture. A general schematic of the forming fixture is shown in Figure 3. More details can be found in a previously published article [18].

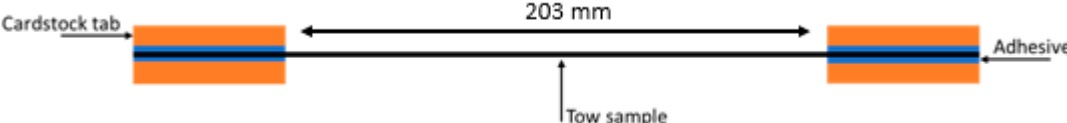

**Figure 2.** Schematic of the fiber reinforced polymer composite tow sample fabrication showing the cardstock tabs adhered to each end of the carbon fiber tows [18].

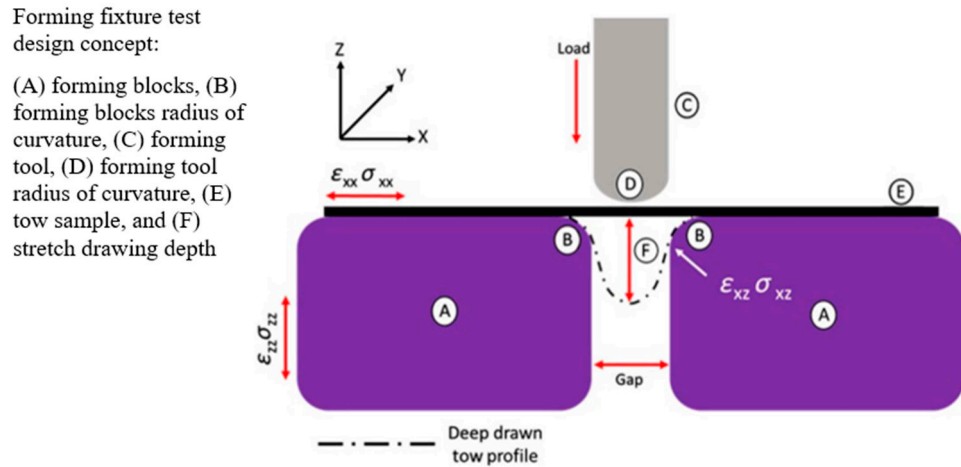

**Figure 3.** Schematic of the MSU designed forming fixture construction [18].

It was hypothesized from the data collected in a previous study that the combination of forming tool diameter (plunger) and gap width could have an interacting relationship that affects the dependent variables [3]. Non-linear models have been reported to correlate numerical and experimental results of CFRP laminated [19]. Therefore, it was essential to study the combination of those factors. A design of experiment (DOE) approach was used to understand the impact of forming tool diameter, gap width, and its interaction on the tensile properties of resinated tows. A Mark-10 (Mark-10 Corp. Copiague, NY, USA) universal test stand and a companion software MESURGauge Plus were used to record the response variables load and displacement. The test data were compiled into MiniTab (Minitab LLC, State College, PA, USA) software for statistical analysis of independent and dependent variables and to develop an ANOVA model for determining the effects of conditional variables on the outputs. This allowed for the identification of factors that influence formability in the stretch-forming test. The statistical significance of these factors provides information on which factors are most significant for formability. ANOVA was conducted at a 95% confidence level for each reaction according to the respective fiber type. A general linear model was employed for the ANOVA with a Box-Cox transformation of $\lambda = 0$ [20].

Radii of the corners of the forming blocks were kept constant at 6.35 mm (0.25 in). Temperature, forming tool diameter, and gap width were identified as important factors that may influence the formability of the CFRP tow samples. For each independent factor, the different levels are shown in Table 4 and Figure 4.

## 2.8. Design of Experiment

A detailed DOE is described in Table 4; it shows the full scope of samples tested and the factors that influence the outputs force (Fmax) and maximum yield displacement (Δmax). Fmax was calculated based on the average of five replicates. Yield displacement refers to the maximum displacement the samples undergo before the force output on the test gauge diminishes. To evaluate formability, the two dependent reaction variables (Fmax and Δmax) and visual observations of the sample's failure mode were recorded. For all tests at RT, except for MSU SBCF, samples failed catastrophically, indicating adverse formability.

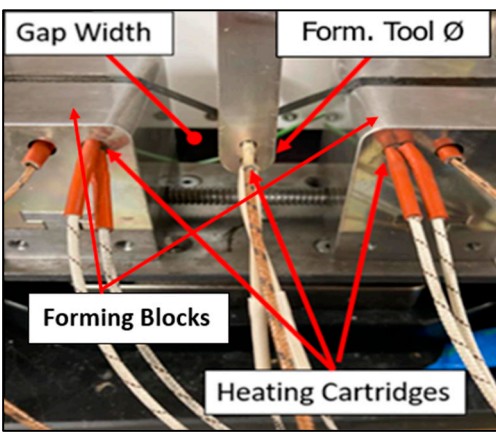

**Figure 4.** Overview of forming fixture and independent variables [17,18].

**Table 4.** List of Independent and Dependent Variables in DOE.

| Independent Variables | | Dependent Variables |
|---|---|---|
| Variables | Levels | |
| Forming tool diameter (mm) | 12.7, 19, 25 | |
| Gap Width (mm) | 16, 28.6, 54 | Maximum yield displacement (Δmax) |
| Forming tool diameter and Gap width Combined (mm) | [12.7, 16], [12.7, 28.6], [12.7, 54], [19, 28.6], [19, 54], [25, 28.6], [25, 54] | Maximum Load (Fmax) |
| Temperature (°C) | 24 ± 1 [Room temperature], 82 ± 1 (Elevated temperature) | |
| Replications | N = 6 | |

## 3. Results and Discussion

### 3.1. Solvent Wash Results

The results from the solvent wash procedure are shown below. Figure 5 shows CFRP samples that have been stripped of the resin matrix, dried, and ready for determinations of its fiber volume.

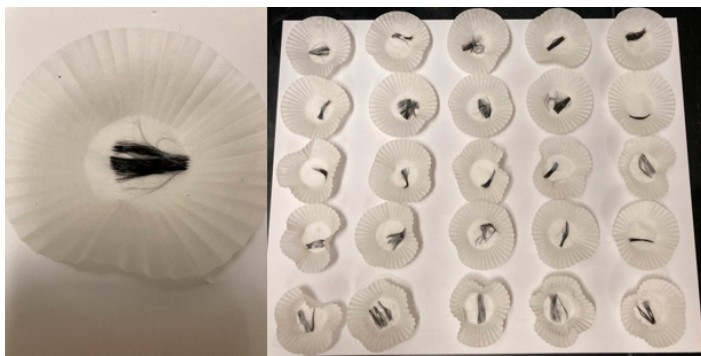

**Figure 5.** Solvent washed CFRP samples.

Ten replicates were tested for each fiber type. Table 5 displays the solvent wash results of the MSU produced pre-preg tows.

The prepreg IM7-GP had the closest fiber volume ratio to what is found in commercial settings. The first generation MSU SBCF showed a fiber volume about 17% lower than IM7-GP. Lower fiber volume could adversely affect the load bearing of the samples because the fiber carries most of the load when CFRPs are stressed.

Response values for Fmax and Δmax for all fiber types; IM7-GP continuous, Hexcel SBCF, and MSU SBCF are displayed in Table 6. Results of analysis of variance (ANOVA)

for average maximum force and maximum yield for three carbon fiber types are displayed in Table 7. Main effect plots are shown in Figures 6 and 7 to illustrate the relative adequate strength of each factor [21].

**Table 5.** Fiber volume ratio of various uncured pre-preg tows.

| Fiber Type | Resin Type | Resin Volume (*n* = 10) | Fiber Volume (*n* = 10) |
|---|---|---|---|
| IM7-GP | RDM2019-053 | 50.1% | 49.9% |
| Hexcel SBCF | RDM2019-053 | 53.6% | 46.4% |
| MSU SBCF | RDM2019-053 | 67.2% | 32.8% |

**Table 6.** Average max force (Fmax), max yield displacement (Δmax), for various CFRP tows.

| | IM7-GP Continuous | | Hexcel SBCF | | MSU SBCF | |
|---|---|---|---|---|---|---|
| | Fmax (N) | Δmax (mm) | Fmax (N) (mm) | Δmax (mm) | Fmax (N) (N) | Δmax (mm) |
| Average | 442.23 | 12.71 | 424.74 | 10.64 | 198.33 | 14.31 |
| Stdev. | 135.06 | 1.56 | 105.14 | 3.58 | 49.23 | 2.15 |

**Table 7.** Results of analysis of variance (ANOVA) for average maximum force and maximum yield for three carbon fiber types.

| | Source | F Value | Contribution (%) | *p*-Value |
|---|---|---|---|---|
| Average maximum force (Fmax) for IM7-GP continuous | | | | |
| | Gap Width (mm) | 27.78 | 38.35% | 0.00 |
| | Forming tool Ø (mm) | 1.19 | 36.36% | 0.31 |
| | Temperature (°C) | 49.59 | 1.26% | 0.00 |
| Average maximum yield disp. (Δmax) for IM7-GP continuous | | | | |
| | Gap Width (mm) | 52.85 | 58.46% | 0.00 |
| | Forming tool Ø (mm) | 1.27 | 1.13% | 0.29 |
| | Temperature (°C) | 26.84 | 11.94% | 0.00 |
| Average maximum force (Fmax) for Hexcel SBCF | | | | |
| | Gap Width (mm) | 41.7 | 3.23% | 0.00 |
| | Forming tool Ø (mm) | 0.02 | 0.00% | 0.99 |
| | Temperature (°C) | 2709.85 | 94.54% | 0.00 |
| Average maximum yield disp. (Δmax) for Hexcel SBCF | | | | |
| | Gap Width (mm) | 35.73 | 30.22% | 0.00 |
| | Forming tool Ø (mm) | 22.6 | 10.36% | 0.00 |
| | Temperature (°C) | 195.14 | 44.74% | 0.00 |
| Average maximum force (Fmax) for MSU SBCF | | | | |
| | Gap Width (mm) | 19.87 | 0.57% | 0.00 |
| | Forming tool Ø (mm) | 3.23 | 0.10% | 0.05 |
| | Temperature (°C) | 6511.04 | 98.37% | 0.00 |
| Average maximum yield disp. (Δmax) for MSU SBCF | | | | |
| | Gap Width (mm) | 72.05 | 19.20% | 0.00 |
| | Forming tool Ø (mm) | 0.93 | 0.23% | 0.40 |
| | Temperature (°C) | 579.09 | 72.55% | 0.00 |

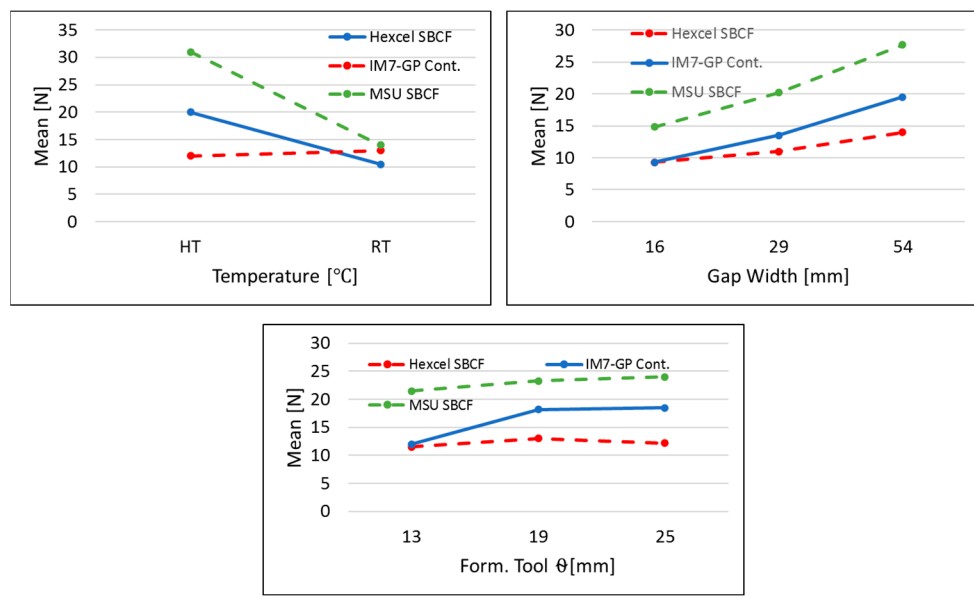

**Figure 6.** Interaction plots of max load (Fmax) for three factors: temperature, gap width, and forming tool diameter.

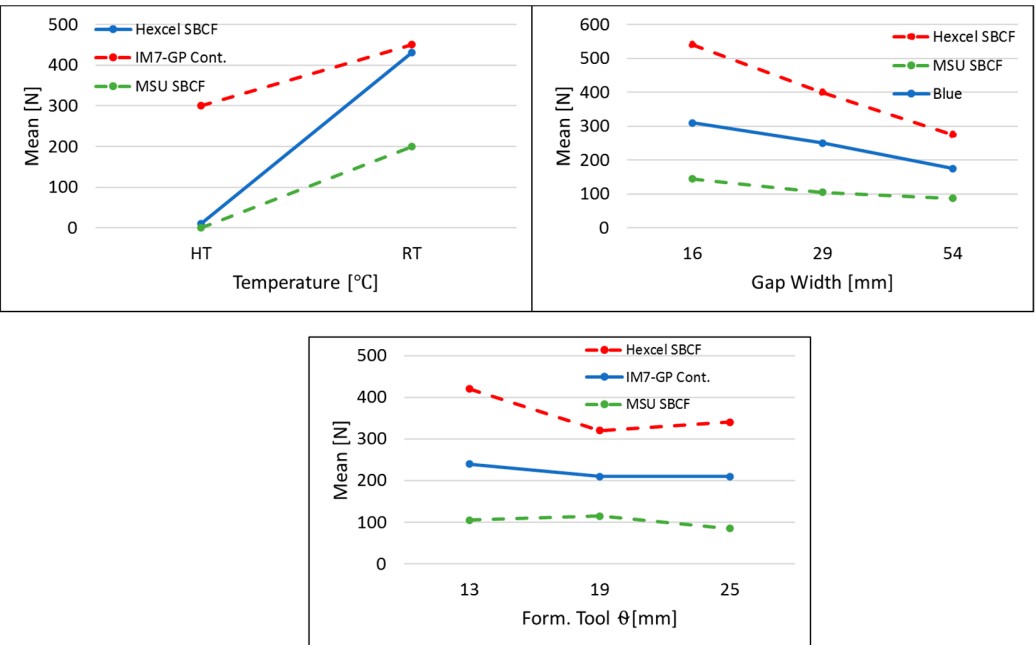

**Figure 7.** Interaction plots of yield displacement (Δmax) for three factors: temperature, gap width, and forming tool diameter.

It is clear from the contribution percentage that temperature has a dominant effect on the Fmax of both the Hexcel SBCF and MSU SBCF, showing 98.37% and 94.54%, respectively. Contribution dependency is the percentage that each factor contributes to the total sequential sums of squares. A higher percentage shows the factor accounts for more of the variation in the response variable. For the Δmax, this effect of temperature is reduced, though still a significant factor in the response. IM7-GP indicates that it is less dependent on temperature, showing a Fmax and Δmax contribution percentage of 1.26% and 11.94% for Fmax and Δmax, respectively. This response is expected since the IM7-GP fibers are continuous throughout the length of the sample despite temperature fluctuations, while the surrounding matrix continues to soften. On the contrary, SBCF are permitted to move freely when subjected to elevated temperatures. The forming tool and gap width effects on

Fmax values of SBCF are not statistically valid enough to show dependency, while the yield displacement values do demonstrate a small contribution. Investigation into the combined contribution of the forming tool and gap widths is required to determine any further effects. The gap width effect on IM7-GP indicates relative high dependency on Fmax of 38.35%. The gap width effect on Δmax for IM7-GP shows a contribution dependency of 58.45%. Mechanically, the fiber is experiencing higher stress from the two directions as the gap width increases, ($\sigma$x and $\sigma$z). When the gap width increases, the force gauge registers lower forces because it records force only in the z-axis. Interaction plots in Figure 6 illustrate the effect of temperature on the SBCF materials while displaying the generally insignificant effect of temperature on IM7-GP fibers.

*p*-values for forming tool diameter in IM7-GP are greater than 0.05, indicating its low significance, while gap width and temperature both show *p*-values less than 0.05, demonstrating that both sources are statistically significant. The average maximum force for Hexcel SBCF has low *p*-values (<0.05) for gap width and temperature, suggesting that both are significant. However, gap width has a minority contribution of 3.23%, while forming tool diameter has a *p*-value of 0.99, pointing toward low significance. All sources of average maximum yield displacement for Hexcel SBCF indicate statistical significance. In addition, MSU SBCF has low *p*-values (<0.05) for temperature and gap width, while forming tool's nose diameter has *p*-values $\leq 0.5$, though the contribution is hardly significant. From the ANOVA model it can be concluded that in general temperature has a significant effect on the Fmax of SBCF materials while it has lower influence on IM7-GP. Alternatively, gap width and forming tool's nose diameter affect Fmax for IM7-GP, where the SBCF materials see little significance from the factors mentioned above.

Similar to Fmax, the ANOVA model indicates that temperature has little effect on the Δmax of IM7-GP but has a larger contribution to the Δmax for both Hexcel SBCF and MSU SBCF. Unlike Fmax though, there is gap width influence on the Δmax of IM7-GP but minor influence from the forming tool nose diameter. For SBCF materials, the contribution of forming tool diameter and gap width is moderately significant, yet still slightly lower than that of the temperature effect.

Lastly, with the individual factors considered, the combination of forming tool's diameter and gap width (interaction) should be considered to determine if they affect the independent variables. Table 8 displays the ANOVA results for all fiber types.

**Table 8.** Results of analysis of variance (ANOVA) for combined forming tool's diameter and gap width (interaction).

| | Source | F Value | Contribution (%) | *p*-Value |
|---|---|---|---|---|
| Average maximum force (Fmax) for IM7-GP continuous | | | | |
| | Forming tool_gap combined (mm) | 15.21 | 43.85% | 0.00 |
| | Temperature (°C) | 54.87 | 26.36% | 0.00 |
| Maximum yield displacement (Δmax) for IM7-GP continuous | | | | |
| | Forming tool_gap combined (mm) | 24.6 | 62.01% | 0.00 |
| | Temperature (°C) | 28.43 | 11.94% | 0.00 |
| Average maximum force (Fmax) for Hexcel SBCF | | | | |

**Table 8.** *Cont.*

| | Source | F Value | Contribution (%) | *p*-Value |
|---|---|---|---|---|
| | Forming tool_gap combined (mm) | 15.3 | 3.26% | 0.00 |
| | Temperature (°C) | 2660.97 | 94.54% | 0.00 |
| Maximum yield displacement (Δmax) for Hexcel SBCF | | | | |
| | Forming tool_gap combined (mm) | 29.04 | 40.75% | 0.00 |
| | Temperature (°C) | 191.29 | 44.74% | 0.00 |
| Average maximum force (Fmax) for MSU SBCF | | | | |
| | Forming tool_gap combined (mm) | 7.7 | 0.70% | 0.00 |
| | Temperature (°C) | 6516.37 | 98.37% | 0.00 |
| Maximum yield displacement (Δmax) for MSU SBCF | | | | |
| | Forming tool_gap combined (mm) | 28.69 | 20.18% | 0.00 |
| | Temperature (°C) | 618.72 | 72.55% | 0.00 |

The results from Table 8 demonstrate that the forming tool's nose diameter and gap combinations do change the results of the ANOVA output. A return of near zero *p*-values for all sources indicates that no individual factor can be completely dismissed. For IM7-GP continuous fiber, the ANOVA trend is similar, as shown in Table 7, in which there is a greater contribution from the geometry as opposed to the temperature. This effect is even greater when considering Δmax, contributing of 62% from the forming tool diameter and gap width combined. The SBCF materials have large contribution percentages from the temperature effects on Fmax of 94.54% for Hexcel SBCF and 98.37% for MSU SBCF. The contribution of the forming tool gap combination on Δmax was much closer, leading to a conclusion that the geometry does affect the Δmax of the SBCF during testing. Figure 8 displays the interaction plots for the ANOVA analysis of forming tool diameter and gap width combined for Fmax and Δmax, respectively. It follows that in general, for continuous fiber, the geometry has a dominant effect on the Fmax while for SBCF the temperature is the most significant contributing factor.

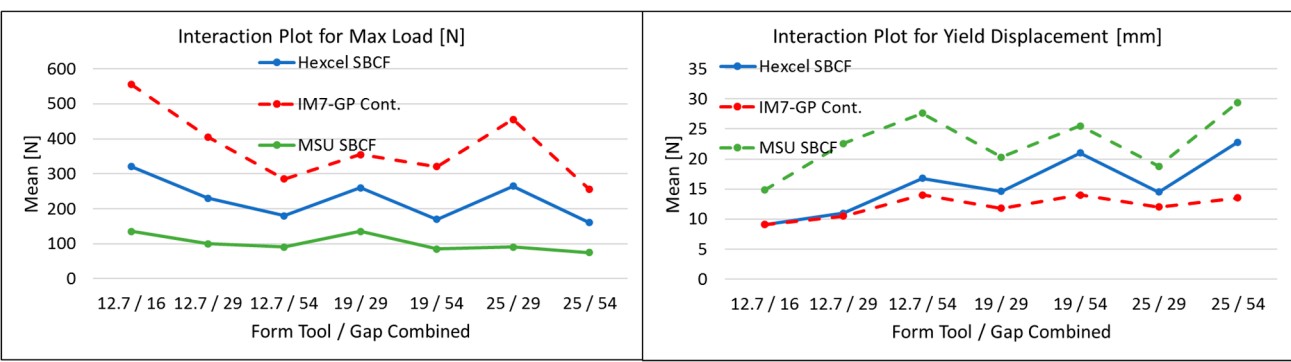

**Figure 8.** Interaction plot of Fmax forming tool and gap combined (A) Interaction plot of Δmax forming tool diameter and gap combined.

### 3.2. Recommendations for Complex Part Design

The general linear ANOVA models can help to determine how CFRPs will form into complex geometries. These models can aid aerospace part designers in understanding the manufacturing limitations that impede the ability for achieving optimized shapes. Data analysis shows in general that SBCF requires less force to form into deep troughs and tight curvatures. In general, for long in-plane structures, traditional continuous CFRPs would still be an acceptable choice, but when bridging, resin pooling and other defects are of concern whereby the superior formability of SBCFs could potentially overcome the production of these flaws [22]. If a fixture with similar geometry as described in this study is used, the force to form the CFRP in the tooling could be estimated, and this would give manufacturing engineers the ability to determine if the production conditions are sufficient for curing the laminate. For example, the force required to form a flat panel with a rib stiffened section running down the middle with a 19 mm wide and 38.1 mm deep trough could be estimated to select the best fiber type for formability. Table 9 shows the regression equations built in MiniTab software and results for IM7-GP continuous CF, Hexcel SBCF, and MSU SBCF to predict the maximum forming load at elevated temperatures based on the geometry described above. It is assumed that the fiber direction is perpendicular to the trough direction, and the forming tool diameter and width are modeled at the same width as the trough to imitate the distributed forces from a vacuum bag.

**Table 9.** Regression equations predicting maximum forming load at elevated temperature.

| | |
|---|---|
| IM7-GP | $Max\ Load[lbf] = 123.5 - 9.1 * \{Forming\ tool\ \varnothing[in]\} - 34.09 * \{Gap\ Width[in]\}$ |
| Hexcel SBCF | $Max\ Load[lbf] = 33.26 - 1.57 * \{forming\ tool\ \varnothing[in]\} - 19.17 * \{Gap\ Width\ [in]\}$ |

## 4. Conclusions

This study investigated the formability aspects of Hexcel continuous IM7-GP, Hexcel SBCF, and MSU SBCF CFRP tows in a stretch-forming fixture using statistical techniques. A general linear ANOVA model was constructed to explore the effect of temperature and fixture geometry (gap width and forming tool diameter) on the response variables Fmax and Δmax. For the IM7-GP continuous CFRP, the ANOVA model gives a strong correlation between Fmax and the geometry of the test fixture. Interaction plots illustrate that as the gap width increases, Fmax decreases, and as the forming tool's nose diameter decreases, Fmax increases. For Δmax, as the gap width increases, Δmax increases, and when the forming tool's nose diameter decreases, Δmax decreases. Overall, with smaller forming tool's nose diameter and small gap width, Fmax should be expected to be higher and Δmax lower. For SBCF materials, temperature was the dominant factor. When considering the geometry of the combined forming tool's nose diameter and gap width there was a slight correlation to the response variable Δmax.

It can be concluded that for continuous fiber CFRP tows forming, Fmax and Δmax are highly dependent on the forming tool's nose diameter and gap width. For continuous fiber CFRP tows forming, Fmax and Δmax do not show statistical significance based on temperature fluctuations from 24 °C to 82 °C. In SBCF CFRP tows forming, Fmax and Δmax are dominated by the temperature during testing in the range of 24 °C to 82 °C, and geometry has a low statistical influence on the Fmax of SBCF CFRP tows. This indicates good formability in shapes consisting of tight radii and deep troughs. Moreover, MSU SBCF CFRP tows show less dependency than Hexcel SBCF on the geometry of the testing fixture for forming force, and contributions of geometry for Hexcel SBCF show 3.26% and MSU SBCF show 0.70%. Finally, lower dependence on tool geometry at higher temperatures indicates possibly superior formability of MSU SBCF.

The results of this study give credence to the difference in formability between the traditional continuous CFRPs and newly developed SBCF CFRPs. Overall, it is shown that SBCF CFRPs have better formability by both measures of Fmax and Δmax, and SBCF fibers developed at MSU demonstrated the highest formability. The higher formability



of the MSU SBCF may result from a better break length distribution when compared to Hexcel SBCF.

**Author Contributions:** Conceptualization, D.S.C. and D.S.B.; methodology, J.C.J. and M.C.E.;. validation, J.C.J., M.C.E., R.A. and C.A.R.; formal analysis, J.C.J. and M.C.E.; investigation, J.C.J., M.C.E., D.S.C., D.S.B., J.C.J. and M.C.E.; writing J.C.J. and M.C.E.; writing—review and editing, D.S.B. and R.A.; visualization, D.S.B., R.A. and C.A.R.; supervision, D.S.B.; project administration, D.S.B. and D.S.C.; funding acquisition, D.S.C. All authors have read and agreed to the published version of the manuscript.

**Funding:** The research was supported by the US Department of the Army under the award number: W911W6-18-C-0050. "The views and conclusions contained in this document are those of the authors and should not be interpreted as necessarily representing the official policies, either expressed or implied, of the Government".

**Data Availability Statement:** The data presented in this study are available on request from the corresponding author. The data are not publicly available due to the proprietary nature of this research.

**Conflicts of Interest:** The authors declare no conflict of interest.

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
