# Peer review of "Design of Experiment to Determine the Effect of the Geometric Variables on Tensile Properties of Carbon Fiber Reinforced Polymer Composites"

_jcs, doi:10.3390/jcs7060222_

Round 1
Reviewer 1 Report
In this work, the authors investigated the influences of the geometric variables on the tensile performance of stretch-broken carbon (SBCF) reinforced polymer composites. This is an interesting topic, taking into account their prevailing importance in the aerospace industry. Although there are only a few insights behind the founds, the results shown in this manuscript look reasonable and practical. Therefore, I am happy to recommend publication, if the issues below can be addressed. Special attention should be paid to some grammar errors and repentances (Page 7, lines 200-203).
Author Response
Response to Reviewers
Ms. Ref. No.: jcs-2259179
Title - Design of Experiment to Determine the Effect of the Geometric Variables on Tensile Properties of Carbon Fiber Reinforced Polymer Composites
Reviewer 1 - In this work, the authors investigated the influences of the geometric variables on the tensile performance of stretch-broken carbon (SBCF) reinforced polymer composites. This is an interesting topic, taking into account their prevailing importance in the aerospace industry. Although there are only a few insights behind the founds, the results shown in this manuscript look reasonable and practical. Therefore, I am happy to recommend publication, if the issues below can be addressed. Special attention should be paid to some grammar errors and repentances (Page 7, lines 200-203).
Response: The authors sincerely appreciate the editor and reviewers for the valuable comments, it helped to improve the quality and content of our manuscript. We have reviewed and revised the language. The changes in the manuscript are marked in red font color.
Reviewer 2 Report
1. Abstract: Stating the combination of fiber and resin studied in this paper seems unnecessary here. The reviewer recommends providing a couple of interesting results from the statistical design of experiments towards the end of the abstract.
2. Line 57: A definition of formability is better placed earlier in the paper, when formability is first touched upon.
3. Line 103: Better context needs to be provided for the meaning of G and GP here.
4. Line 110: This paragraph is tough to digest. The reviewer recommends placing this information in a table since the same variables are discussed for all 3 different types of CF. Suggestions for columns are break length, manufacturer, sizing, and testing agency/mechanism; but the authors are better positioned to format the table.
5. Line 134: Please use subscripts when denoting volume ratios and volumes. This is fairly straightforward in both word and latex.
6. Line 137: If there is manual, publication or chemical/physical explanation touching upon action of the solvent here, it would be a good addition here.
7. Line 176: Maximum yield displacement ( max)?
8. Line 179: What do the authors mean by visual failure mode?
9. Line 157: The authors cannot refer the entire experimental design to another publication. At least, an abridged version of the experimental design has to be presented in this paper. This is especially pertinent since the main thesis of the paper is centered on a DOE approach using data from tests conducted using the forming fixture referred to here.
10. Line 158: The authors have not explained what the forming tool diameter or gap width means. The discussion in this section is meaningless for a reader unless proper context is provided.
11. Line 183: Forming block? First time this is mentioned in the paper.
12. Line 157: Why was this hypothesis made? Was there a physical reason or data driven reason for the authors to speculate this?
13. Line 183: Temperature has been added to the list of variables from earlier. Why the addition here?
14. Line 200: Paragraph is repeated.
15. Table 6: Units?
16. Table 7: How many tow samples were tested for each of the levels shown in table 3? Is the data provided in the paper? Can the authors comment on the repeatability of their tensile experiments? Was there an effort to check for it?
17. Line 231: Nose diameter?
18. Line 211: The discussion of results starting from this point is incomprehensible unless the reader has had the opportunity to appreciate what the authors mean by gap width and tool diameter, which has not been done by the authors.
19. The interaction plots in page 11 is hard to follow. The reviewer recommends splitting the comparisons up and presenting results while keeping one variable constant.
20. Line 274 -290: The discussion here is good and needs to present earlier in the paper in the introduction section to better motivate the paper.
21. Table 9: How are the regression equations here generated?
Author Response
Ms. Ref. No.: jcs-2259179
Title - Design of Experiment to Determine the Effect of the Geometric Variables on Tensile Properties of Carbon Fiber Reinforced Polymer Composites
Reviewer 2 - Abstract: Stating the combination of fiber and resin studied in this paper seems unnecessary here. The reviewer recommends providing a couple of interesting results from the statistical design of experiments towards the end of the abstract.
Response: The authors sincerely appreciate the time and efforts put by the reviewer and the valuable comments. It helped to improve the quality and content of our manuscript. We have reviewed and revised the language in the abstract and added additional results. The changes in the manuscript are marked in red font color.
- Line 57: A definition of formability is better placed earlier in the paper, when formability is first touched upon.
- Comment addressed, manuscript revised
- Line 103: Better context needs to be provided for the meaning of G and GP here.
- Comment addressed additional information added, however the proprietary formulation are difficult to describe as the exact formulation is not disclosed to public.
- Line 110: This paragraph is tough to digest. The reviewer recommends placing this information in a table since the same variables are discussed for all 3 different types of CF. Suggestions for columns are break length, manufacturer, sizing, and testing agency/mechanism; but the authors are better positioned to format the table.
- Comment addressed, manuscript revised
- Line 134: Please use subscripts when denoting volume ratios and volumes. This is fairly straightforward in both word and latex.
- Comment addressed, manuscript revised
- Line 137: If there is manual, publication or chemical/physical explanation touching upon action of the solvent here, it would be a good addition here.
- Comment addressed, manuscript revised
- Line 176: Maximum yield displacement ( max)?
- Comment addressed, manuscript revised
- Line 179: What do the authors mean by visual failure mode?
- Comment addressed, manuscript revised
- Line 157: The authors cannot refer the entire experimental design to another publication. At least, an abridged version of the experimental design has to be presented in this paper. This is especially pertinent since the main thesis of the paper is centered on a DOE approach using data from tests conducted using the forming fixture referred to here.
- Comment addressed. Authors have added additional information and a new figure to describe the forming fixture.
- Line 158: The authors have not explained what the forming tool diameter or gap width means. The discussion in this section is meaningless for a reader unless proper context is provided.
- Comment addressed, manuscript revised
- Line 183: Forming block? First time this is mentioned in the paper.
- Comment addressed, manuscript revised
- Line 157: Why was this hypothesis made? Was there a physical reason or data driven reason for the authors to speculate this?
- Comment addressed, manuscript revised
- Line 183: Temperature has been added to the list of variables from earlier. Why the addition here? Thanks for the comment but we not sure what this means
- Line 200: Paragraph is repeated.
- Comment addressed, manuscript revised
- Table 6: Units?
- Comment addressed, manuscript revised
- Table 7: How many tow samples were tested for each of the levels shown in table 3? Is the data provided in the paper? Can the authors comment on the repeatability of their tensile experiments? Was there an effort to check for it?
Thank you for the suggestion. Authors have added the number of replications to the table 3. Six Replication added to table 3. Repeatability was validated from prior testing and the paper on the forming fixture
- Line 231: Nose diameter
- Comment addressed, manuscript revised
- Line 211: The discussion of results starting from this point is incomprehensible unless the reader has had the opportunity to appreciate what the authors mean by gap width and tool diameter, which has not been done by the authors.
- Author have addressed this comment by adding an new figure.
- The interaction plots in page 11 is hard to follow. The reviewer recommends splitting the comparisons up and presenting results while keeping one variable constant.
-Authors have improved the quality of the figures, but we are limited by the capability of the software.
- Table 9: How are the regression equations here generated?
- Comment addressed, manuscript revised
Author Response
Ms. Ref. No.: jcs-2259179
Title - Design of Experiment to Determine the Effect of the Geometric Variables on Tensile Properties of Carbon Fiber Reinforced Polymer Composites
Reviewer 3 - The manuscript entitled: Design of Experiment to Determine the Effect of the Geometric Variables on
Tensile Properties of Carbon Fiber Reinforced Polymer Composites presents interesting research
results.
The authors of the manuscript to study and validate the performance of SBCF (Stretch Broken
Carbon Fiber) a statistical designed of experiment was conducted using three different types of
CFRPs (Carbon Fiber Reinforced Polymers) in tow/tape form. The results presented in this paper can
help to define useful testing methods to compare different CFRPs as well as provide a repeatable
approach to creating a statistical model for measuring results from the formability trials.
The research results described in the article correspond to the topics of the journal Journal of
Composites Science. The following comments will help to improve the manuscript:
The authors sincerely appreciate the time and efforts put by the reviewer and the valuable comments. It helped to improve the quality and content of our manuscript. We have reviewed and revised the language, added new reference and additional results. The changes in the manuscript are marked in red font color.
1) In my opinion, the introduction to the topic could be expanded. Manuscripts presenting
experimental studies of reinforced polymer composites could be helpful. I suggest reading the
following papers presenting studies of the behaviour of construction made of CFRP.
The authors have added a new reference suggested to the manuscript - Non-linear analysis of the postbuckling behaviour of eccentrically compressed composite channel-section columns.
2) To which Table 1.3 refers in the following sentence:
“The mechanical properties of continuous Hexcel IM7 CF bundled in a 12,000 fiber (12K) tow are
shown in Table 1.3.”
- Comment addressed, manuscript revised
3) What method was used to make the real composite objects? Please state the parameters of the
manufacturing process.
This comment has been covered in the CF sample preparation section
4) For a better understanding, please add in more figures: the figure of the actual sample, the
experimental position with the sample is missing....
The authors have added a new figure that would help the readers.
5) A presentation and description of the test rig is recommended. Please include drawings of your
experimental studies. Describe in detail the experiment carried out based on the figures. Real figures
of punched tests, tensile tests and bearing tests...
Thank you for the suggestion. The authors have added a new figure that clearly shows the test fixture and important variables.
Round 2
Reviewer 1 Report
The authors have addressed all comments and the revised manuscript is acceptable for publication.
Author Response
The authors want to thank you for your time and effort involved in reviewing our manuscript. We appreciate your input.
Reviewer 2 Report
1. The authors have not addressed the point regarding stating the combination of fiber and resin in the abstract. The reviewer continues to feel that this is unnecessary. If the authors disagree with this, please provide an explanation.
2. In what way has comment 11 been addressed?
3. Can the authors please point out the locations where the changes have been made? Either please provide line numbers or please provide the revised content in a point-by-point format. For example, the reviewer cannot recall what comment 12 pertains to. It is up to the authors to provide a reminder of the original content from this section OR provide the revised content OR at least point to the location of the changes. Please provide this.
Author Response
The authors appreciate your time and effort involved in reviewing our manuscript. The comments helped to improve the quality and content of our manuscript. We have addressed your three comments to the best of our ability and hope this will address your concerns (all changes are marked in red).
- The authors have not addressed the point regarding stating the combination of fiber and resin in the abstract. The reviewer continues to feel that this is unnecessary. If the authors disagree with this, please provide an explanation.
Thank for the suggestion. Authors have revised the abstract by adding a section on the results. See page 1, line 27-33.
- In what way has comment 11 been addressed?
The comment 11 has been addressed by revising the figure 3A (pg5 line162-164) and figure 3b (pg7 line186). The figure 3A description shows (A) Forming blocks. Similarly, in figure3B, a new label has been added to describe the forming block.
- Can the authors please point out the locations where the changes have been made? Either please provide line numbers or please provide the revised content in a point-by-point format. For example, the reviewer cannot recall what comment 12 pertains to. It is up to the authors to provide a reminder of the original content from this section OR provide the revised content OR at least point to the location of the changes. Please provide this.
Pg. 2 line 74-75. Based on our previous knowledge and work published in 2021 we hypothesized that this statistical modelling would help in determining the forming forces. Therefore, a design of experiment (DOE) approach was used to understand the impact of forming tool diameter, gap width and its interaction (independent variables) on the tensile properties (dependent variable) of resinated tows.
Reference: J. Janicki, M. Egloff, R. Amendola, C. Ryan, D. Bajwa, A. Dynkin and D. Cairns, "Formability Characterization of Fiber Reinforced Polymer Composites Using a Novel Test Method," Journal of Testing and Evaluation, vol. 50, no. 2, 2021.
Reviewer 3 Report
Accept in present form
Author Response

(The authors gave the same response as above.)

Round 3
Reviewer 2 Report
N/A